# Cover Crop Yield, Nutrient Storage and Release under Different Cropping Technologies in the Sustainable Agrosystems

**DOI:** 10.3390/plants12162966

**Published:** 2023-08-16

**Authors:** Aušra Arlauskienė, Lina Šarūnaitė

**Affiliations:** 1Joniskelis Experimental Station, Lithuanian Research Centre for Agriculture and Forestry, 39301 Joniskelis, Lithuania; ausra.arlauskiene@lammc.lt; 2Institute of Agriculture, Lithuanian Research Centre for Agriculture and Forestry, Instituto 1, 58344 Akademia, Lithuania

**Keywords:** biomass NPK, clay loam, C:N, red clover, soil mineral N, white mustard

## Abstract

Due to short post-harvest seasons, it is not always possible to grow worthy cover crops (CCs). This research aims to clarify the impact of undersown red clover (*Trifolium pratense* L., RC) and post-sown white mustard (*Sinapis alba* L., WM) management on their biomass, accumulated nitrogen (N), phosphorus (P), and potassium (K) content and the nutrient release to subsequent main crops. During the study period, RC mass yields varied from 220 to 6590 kg ha^−1^ DM and those of WM from 210 to 5119 kg ha^−1^ DM. WM shoot biomass increased with the increase in rainfall in August and the average daily temperature of the post-harvest period. CC productivity and efficiency were higher when growing short-season spring barley than winter wheat. In the warm and rainy post-harvest period, undersown WM after winter wheat increased the biomass by 34.1% compared to post-harvest sowing. The application of straw (+N) increased the accumulation of nutrients in WM biomass. The intensive fertilization of the main crop had a negative effect on RC yield and NPK accumulation. RC shoot biomass was characterized by a higher N content and WM by a higher P concentration. Well-developed CCs could reduce soil mineral nitrogen content by 28.5–58.8% compared to a plot without CCs. Nutrient transfer to spring barley was dependent on the N content of CC biomass and the carbon and nitrogen ratio (C:N < 20). We conclude that CC growth and efficiency were enhanced by the investigated measures, and in interaction with meteorological conditions.

## 1. Introduction

Biodiversity lost, soil degradation, excess nitrogen in water and soil systems, increasing carbon dioxide content in the atmosphere—these are the consequences of short crop rotations, high N fertilizer rates, intensive tillage, and heavy use of pesticides [1]. In the soil, the composition and intensity of the activity of microorganisms, the biochemical processes causing the intensity of mineralization processes of organic substances and soil degradation, the emission of greenhouse gases, and undesirable phenomena are changing [2]. The instability of crop yield and quality indicators in an unpredictable economic environment is cause for concern. These problems are exacerbated by climate variability and change [3]. When agricultural practices are not carried out on time, the tools used work with incomplete efficiency, the cost of production increases, and competitiveness decreases [4]. In Lithuania, main crops occupy arable land only during 60–70% of the warm season, when positive temperatures are above +5 °C. The remaining 30–40% of the warm season (about 60–80 days) is, therefore, used unproductively, leading to the physical and chemical degradation of the soil [5]. Due to climate change, dry periods and uneven distribution of rainfall during the growing season are leading to higher daily average temperatures and longer growing seasons without plants [3]. This promotes microbial mineralization of soil biomass and N release between September and November (and in some cases in December). Large N amounts are lost, posing risks to air, water, soil, and biodiversity. N leaching is highest in light soils, but also in heavy soils, as due to their increased coherence, vertical cracks during the growing season and increased runoff from ploughed soils in winter create conditions for N to migrate to deeper layers and pollute groundwater [6].

Cover crops (CCs) are an important component of sustainable agricultural production. Cover crops are an important component of sustainable agricultural production. According to Blanco-Canqui [7], the term CCs is defined as a close-growing crop that provides soil protection, seeding protection, and soil improvement between periods of main crop production. Many reviews indicate that CCs are multifunctional [7,8].

In addition, CCs in the post-harvest period reduce evaporation and ammonium nitrogen (N) emission into the environment [9]. Cover crops can significantly reduce N leaching [10] and net greenhouse gas balance and increase soil organic carbon (SOC) sequestration [9,11]. Increases in SOC are directly linked to improved soil quality and adaptation to climate change [12]. During the autumn period, the constant maintenance of the soil surface with CCs protects it from the further negative effects of direct weathering. Cover crops alleviate soil compaction, improve soil structural and hydraulic properties, moderate soil temperature, and suppress weeds [7,8]. The surface of heavy loam soils without CCs becomes slushy and, as they dry, vertical cracks form, increasing the risk of nutrients being leached into groundwater [6]. Cover crops can increase water infiltration [13,14], which is relevant in reduced tillage or direct drilling systems [15]. In general, CCs have significantly increased parameters of soil microbial abundance, activity, and diversity [2]. Soil microorganisms, in turn, release inorganic nutrients through SOM mineralization, stimulate the soil enzymatic complex, influence plant nutrient acquisition, and drive nutrient cycling [16]. One of the potential disadvantages of CCs is the reduction in the yield of the subsequent main crops [9,17,18]. Garba et al. [18] reported that cover crops reduced the subsequent crop yield by 7%, soil water content by 18%, and soil mineral N by 25%, with significant variation across climates, soil types, and crop management conditions. Despite several limitations, CCs improve overall soil health and ensure environmental sustainability [12,19].

In Lithuania, it is not possible to grow a post-harvest cover crop every year. On the one hand, the biomass yield of CC plants varied due to the predictable effects of extreme weather events and long-term climate change, while on the other hand, the increase in temperature with climate change prolonged the growing season suitable for CC production. The duration and intensity of light during this period, the sum of positive temperatures, the diurnal variation in temperature, and the amount and distribution of rainfall have a significant impact on post-sown CC germination and the above-ground mass formation [6]. The Lithuanian climate is more favourable for the development and biomass accumulation of undersown CCs, as they have already passed the first stages of development after the main crop harvesting and are, therefore, always more likely to cover the soil surface more quickly and to produce a larger biomass. However, these CC plants are mostly grown on organic or sustainable farms, whereas on intensive farms, the undersown crop often suffers from the heavy fertilization of the main crop. Many of the studies and recommendations on CC cultivation have been conducted and applied in warmer climates. However, water scarcity, short growing seasons, and repeated freeze–thaw cycles have been cited as limiting factors for CC production. Cover crops are prone to technological problems, including the method of killing the host for pathogens, and regeneration [19]. The effect of CCs was less pronounced under conditions such as continental climate, chemical CC termination, and conservation tillage [2]. Although the economic interest of these crops is low, their main objective is to protect the soil and nutrients. In Lithuania, the harvest productivity of the CCs is directly related to the accumulation of nutrients in them.

Studies have shown that many of the disadvantages can be avoided by optimising CC cultivation technologies. One strategy to avoid subsequent crop losses is to select CCs in mixtures of legumes and non-legumes [9,20], and to improve CC management to increase nutrient availability [21]. Researchers have found that the mass and nutrient composition of CC plants vary considerably between plant species [10,21] and between different parts of the same plant [22]. *Vicia villosa* Roth has been found to be a good accumulator of N, *Raphanus sativus* L.—of S, P, and K, and *Lupinus albus* L.—of N, P, and Mn [21]. Therefore, by growing mixtures of different cover crops, it is possible to achieve accumulation of more varied nutrients in CC biomass, and synchronisation of their release with the main crop demand [21,23,24]. The release of nutrients from CC biomass depends on its biomass and chemical composition [21,22] as well as the tillage method [25,26,27]. In most cases, the effectiveness of CCs can only be observed after long-term use [25]. Sharma et al. [19] argued that only the long-term use of CCs leads to economic benefits.

Chalise et al. [14] argued that CC efficiency varies considerably and depends on the region and cropping system. Agronomic practices related to CCs should therefore be adapted to the specific soil and climatic conditions of the region [9]. We hypothesize that the optimization of CC technologies, taking into account the applied agronomic practices, can partially reduce the negative influence of meteorological conditions and increase the efficiency of CCs.

This study aims to elucidate cover crop species and their sowing method (undersown and post-sown), as well as crops grown as the main crops, their fertilization intensity, the use of straw for fertilizer, the impact on the CC biomass, accumulated N, P, K, and their release and transfer to a subsequent crop.

## 2. Results and Discussion

### 2.1. The Influence of Cover Crop Types, Their Sowing Methods, and the Fertilization of Main Cropss and the Use of Straw as Fertilizer on Cover Biomass and Accumulated Nutrient Content

Cover crop species and their functions (Experiment I). The experiments showed that CC biomass depends on the interaction between plant species (main crops and cover crops) and the meteorological conditions of the growing season (assessed through experiments, *p* < 0.01) (Table 1). The comparison of CC shoot biomass in different experiments showed that RC mass yields varied from 220 to 6590 kg ha^−1^ DM t and those of WM from 210 to 4580 kg ha^−1^ DM.

The highest coefficient of variation (V) was determined when growing WM and RC after winter wheat in the field from 31.6 to 42.5% (except for Experiment Ia). Clover produced a 1.4–2.9 times higher yield compared to WM (except in Experiment Ib). White mustard is nitrogen intensive, thus being able to take up nitrogen from the soil in the post-harvest period. As red clover is more valued for its biological N fixation [7], it is not efficient in N uptake from the soil. In the years favourable (Experiment Ia) to CC growth, RC accumulated on average 68.3 kg N ha^−1^ more compared to WM. The amount of phosphorus depended significantly on the interaction between the experiment and CCs. The most P was accumulated in the RC crop in Experiments Ia and Ic. The K content was influenced by the interaction of all three factors. Growing WM in the spring barley field resulted in a higher aboveground mass and N and K accumulation in most cases compared to growing it in the winter wheat field. This was due to the longer growing period of wheat and the higher N demand compared to those of spring barley. In less favourable growing years, CCs demonstrated a low aboveground mass and low nutrient accumulation, which did not differ significantly between the plant species (Experiment Ib). In most cases, the high N content of legume biomass resulted in a low C:N ratio (9–14), leading to the faster mineralisation of incorporated organic matter (15–29). At the same time, that indicator for WM was wider (15–29).

Cover crop sowing methods and straw (Experiment II). Teixeira et al. [28] indicated that the main determinant of variability in CC biomass is the sowing date, followed by weather and soil type. White mustard reacts positively to the duration and intensity of light, so an earlier sowing method was chosen—we scattered the seeds at the waxy maturity stage of winter wheat (2 weeks before the winter wheat harvest). Cover crops and their sowing methods had a significant effect on plant biomass, but in relation to the year of the experiment (*p* < 0.01) (Table 2).

According to the average data, the use of straw as fertilizer increased CC biomass by 8.4% (*p* < 0.05). The evaluation of WM sowing methods showed that, in the less favourable year (Experiment IIa), the WMus biomass varied (V = 45.3%) and was significantly lower than WMps. The formation of a higher RC biomass was hampered by the dry period during seed germination, which lasted for the whole month of May. In 2006, a favourable year for CC growth with a long, warm, and rainy post-harvest period, the WMus mass was 4.5–4.9 times and that of WMps 2.4–3.2 times higher than those in 2004. WMus sown early were exposed to better germination conditions on the shaded soil surface by winter cereals, resulting in a substantially higher yield of WMus biomass (34.1% on average) compared to those sown immediately after cereal harvesting. The yield of post-sown NLps was 1.7–1.8 times lower than that of WMps. The amount of N and K accumulated in the biomass of cover crops was significantly influenced by the interaction of all studied factors: the amount of P and the interaction between the year of the experiment and CCs (*p* < 0.01). The undersown WMus accumulated the highest N, P, and K contents in biomass, which were significantly higher than those accumulated in the biomass of WMps (except WS) or NLps plants. The value of the C:N ratio was dependent on the meteorological conditions and showed little difference between RC and WM (10–15).

Fertilization systems (Experiment III). On a typical arable farm, the only use of straw is for fertilization. However, when cereal straw is used as fertilizer, there is often competition between CC plants and micro-organisms for nitrogen and moisture in the soil [19]. Studies have shown that biomass is significantly influenced by the interaction between CCs and straw use and between CCs and main crop fertilization (*p* < 0.01) (Table 3).

The experiments showed that, in unfertilized plots, RC produced the highest shoot biomass, on average 3.4 times higher than that of WM. In spring barley, the use of SF and IF reduced RC biomass by an average of 3.2 and 3.5 times, respectively, compared to the unfertilized plots. Straw application also reduced RC shoot biomass (significantly in fertilized plots). However, straw application (+N) had the largest positive effect on WM biomass. Compared to WM biomass, statistically significant N concentration increases of 1.33 and 1.79 g kg^−1^ on average were found in RC biomass (WS and S, respectively). The application of straw as a fertilizer decreased the N concentration in WM biomass and increased it in RC biomass. However, WM accumulated significantly more P per unit of biomass by an average of 0.16 g kg^−1^ and 0.11 g kg^−1^ (WS and S, respectively) compared to RC. The potassium concentration in WM and RC biomass did not differ but was the highest in IF variants (such as N). Nutrient accumulation in biomass was significantly dependent on the interaction between CCs and straw and CCs and main crop fertilization (*p* < 0.01) (Figure 1). The highest nutrient accumulation was in RC biomass (UF). The fertilization of the main crop did not affect the nutrient accumulation in WM biomass (WS). However, the application of straw as fertilizer increased the nutrient accumulation (SF and IF) and was, in most cases, equal to that of RC biomass. Fertilization had an effect on the mineralisation rate (C:N) of WM biomass, which decreased as fertilization increased. The C:N ratio of RC was generally lower (on average, 15) and showed a more rapid mineralisation of biomass than that of WM.

Research has shown that CC biomass is influenced by plant species, sowing methods, use of straw as fertilizer, and main crop plants and their fertilization, but in interaction with the meteorological conditions of the experiments. The data from the three experiments showed a greater response of post-sown rather than undersown CC plants to meteorological conditions. Dry periods during intensive crop growth (May) or maturity (second half of July) resulted in higher WM yields. That was due to the under-utilisation of nutrients by the main crop (Table 2, Experiment IIb). If the harvest of the main crop is delayed, the potential for producing a more valuable CC biomass is reduced. Our research showed the dependence of WM shoot biomass on the amount of rainfall in August and the average daily temperature of the post-harvest period; with an increase in these parameters, WM biomass increased (Figure 2).

The success of undersown legumes depends on the correct seed rate of grasses and cereals, and on the moderate N fertilization of cereals. A too-dense cereal crop and a high N fertilizer rate can severely suppress RC growth. Red clover was the most sensitive to drought during germination. Our data show that a spring barley yield of 6 t ha^−1^ can completely destroy the undersown RC (Table 2, Experiment IIb). The above-mentioned reasons determined differences of up to several times in CC shoot biomass. The germination, growth, and nutrient accumulation of WM are enhanced by earlier dates of sowing (WMus). The amount of post-sowing rainfall has a positive effect on WM germination and CC biomass production [7]. Post-sowed CCs are worth cultivating after crops with a shorter growing season, as they take fewer nutrients from the soil. SMN and its availability are of great importance for the mass of non-legume plants [27,29]. After harvesting, the higher content of SMN is determined by over-fertilization with mineral and organic fertilizers, late fertilizer application [4], and previous cropping [26]. Our experiments showed that to reduce competition between WM and micro-organisms and to increase CC yields and nutrient accumulation, it was worth applying a starter N rate with straw. This was also observed by other researchers [30].

### 2.2. Influence of Cover Crops on Soil Mineral N Variation in Autumn and Spring

Cover crop sowing methods and straw (Experiment II). The research showed that, in many cases, WM was better at capturing SMN from the soil than RC. The efficiency of WM depended on its shoot biomass [31]. In less favourable growing years, CC plants produced shoot biomass, which had no significant effect on the SMN content (0–80 cm layer) (Figure 3a).

The data show that the WM biomass of 508–1402 kg DM ha^−1^ and RC biomass of 1814–2406 kg DM ha^−1^ were not effective in capturing free N from the soil [32]. In a favourable growing year, post-sown CCs reduced the amount of SMN by 51.5–58.8% compared to the plots without CCs [31]. Other researchers have reported that CCs reduce N leaching by 36 to 62% [25] or 56% [10]. The statistical analysis showed that the increase in CC shoot biomass from 1892 to 5313 kg DM ha^−1^ was accompanied by a significant decrease in SMN content in the 0–80 cm soil layer (Figure 3b). Thapa et al. [10] suggested that N leaching is reduced by non-legume CCs at masses between 4 and 8 kg DM ha^−1^. It is worth noting that RC grown under favourable conditions can reduce SMN content similarly to poorly developed WM [33].

Fertilization systems (Experiment III). The cultivation of RC and WM cover crops during the autumn period resulted in a decrease in SMN content (by 23.5 and 28.5%, respectively) compared to unfertilized plots that did not have CCs or receive straw additions [33]. In the spring of 2014 (after CC biomass incorporation), the SMN contents in the soil were significantly influenced by the use of WM as a cover crop (*p* < 0.01), the utilisation of spring barley straw (*p* < 0.01) from the previous year, and the intensity of barley fertilization (*p* < 0.05). The use of the previous year’s spring barley straw (+N) significantly increased the content of SMN by 13.8% on average, compared to the soil without straw. In the treatment with WM, the use of mineral fertilizers reduced the content of SMN, while after applying sustainable fertilizing, the decrease was significant [34]. This is likely due to microbial N immobilisation [23]. In spring, the SMN content showed a moderate positive correlation with CC biomass and N concentration (Figure 4a,b). Soil mineral N showed a negative correlation with CC shoot biomass C:N (Figure 4c).

This indicates that, in spring, SMN content depended on the applied CC mass, its N content, and mineralization intensity (C:N). It is shown that the most yield benefits were proportionate to the soil water content and SMN at the time of the sowing of the subsequent crop [18].

Our research is in agreement with the findings of other researchers, which showed that the earlier sowing that increases shoot biomass increased the growth duration of CCs and reduced the SMN content in autumn [10,28]. The subsequent important step is to retain the collected nutrients until the beginning of the next growing season. When the CC mass is incorporated in autumn, mineralisation begins. Many CC legumes are a good source of N storage, but due to their narrow C:N ratio and low lignin content (<3%), the destruction of the organic matter they accumulate is in the direction of mineralisation and the risk of leaching of nutrients during winter remains [23,35,36]. These studies did not take into account the underground plant mass. Legume shoot biomass is incorporated into plant roots, which, due to a higher lignin content and the specific location of tissues (creating barriers to the entry of decomposer communities), slow down the mineralisation of the total biomass to some extent [36].

Unlike RC, WM biomass transformation can proceed in several directions: either to degrade and release N or to promote microbial N immobilisation. In our studies, the C:N ratio of WM biomass varied over a wide range from 10 to 29. This ratio was influenced by the stage of WM development, that is, the ratio of readily and poorly degradable compounds (leaves and stems). When exposed to less favourable growing conditions, WM forms a lower aboveground mass, develops more rapidly, and its biomass C:N is expanded. Without limiting WM growth, i.e., without interrupting its vegetation, the plants mature by forming pods and the stems develop tissues resistant to destruction. This often results in a wider range of C:N (>20) and a temporary microbial N immobilisation [36]. The rate and direction of plant biomass decomposition depend not only on the biomass and its quality, but also on the yield, soil texture, physico-chemical and biological properties, and meteorological conditions [21]. Lawson et al. [20] suggest that CC plant mixtures provided a balance between biomass accumulation, N concentration, and mineralisation intensity.

During the autumn and winter periods, an effective means of inhibiting biomass mineralisation and nutrient migration to deeper layers is the incorporation of CC nitrogenous biomass together with the carbon-rich straw of cereals. This practice not only reduces SMN content but also increases the formation of stable humic substances and improves aeration [14,37]. Agronomic practices, such as delaying CC biomass application, winter intercropping, using live and dead mulch technologies, and the application of no-till systems, contribute to reducing the mineralisation of incorporated biomass [25,27].

### 2.3. Nutrient Release from CC Shoot Biomass to Subsequent Crops

Cover crop species and their functions (Experiment I). After CC biomass incorporation, nutrients can be offered to the subsequent crop, added to the soil organic matter or released as losses to the environment. The effect of CC biomass on the nutrient content of spring barley yields (grain and straw) is presented in Table 4. The accumulation of N in spring barley yields due to CCs was not consistent: RC increased biomass in two of the years, and WM in two of the three years. When considering each experiment separately, WM intercalation with spring barley combination increased the accumulated N by an average of 7.1 kg ha^−1^ or 8.3% compared to WCCs. The incorporation of RC mass significantly increased the N content of spring barley yield by 4.2–12.5 kg N ha^−1^ or 5.3–12.1% compared to WM biomass (except for the data from Experiment Ib). In Experiment Ib, the incorporation of a low CC mass had no significant effect on the spring barley yield. According to other researchers, the mass of CCs < 2 t ha^−1^ does not perform its functions [38]. Growing CCs in the spring barley field resulted in a higher N accumulation in yield than growing CCs in the winter wheat field. Cover crop RC and WM mass increased P accumulation in spring barley yields (in most cases significantly) and were almost equivalent in their effectiveness. Potassium accumulation in barley yields was similar to that of N. However, significant differences compared to the control plots were found only after the incorporation of RC mass (an increase of 4.6–8.4 kg K ha^−1^ or 4.7–9.8%).

Cover crop sowing methods and straw (Experiment II). The research showed that N accumulation in the spring barley yield was influenced by the experiment year, cover crops (*p* < 0.01), and straw use (*p* < 0.05), and phosphorus accumulation was influenced by the interaction between the experiment and straw (*p* < 0.05). K accumulation in the crop was influenced by the interaction of all the factors studied (*p* < 0.01) (Table 5). According to the average data, in Experiment IIa, N accumulated in the spring barley crop was on average 19.1 kg ha^−1^ or 24.4% less compared to data from Experiment IIb. In both trials, the application of straw as a fertilizer also reduced N accumulation in the barley crop.

In both experiments, N accumulation in the barley crop was significantly increased by WM (except for Experiment IIa, where straw was used as a fertilizer). The legumes had an increase of 3.7 kg N ha^−1^, WMps 7.3, and WMus 6.6 kg N ha^−1^ (on average), compared to the plot without CCs. The results in this experiment were due to the fact that WM produced a significantly higher mass than the legumes (especially NL in Experiment IIb), and all CCs had a similar C:N biomass ratio (<15).

The CCs in the barley yield resulted in a 0.3–1.2 kg P ha^−1^ increase (WS). The P content accumulated in spring barley yields differed among the experiments: 16.5 kg P ha^−1^ in Experiment IIb and 11.7 kg P ha^−1^ (on average) in Experiment IIa. The various mechanisms of P uptake have been discussed: P transfer via cover crop residues, organic anion exudates, root-exuded enzymes, and microbial interactions [39].

The highest K accumulation was observed in barley grown after legume CCs, with a difference of 13.0–22.1% compared to the plots without CCs (especially in plot WS). The post-sowing method of WM had a significant effect on K accumulation in the barley yield, compared to WMus (except the WS Experiment IIb). White mustard was post-sown in the soil with shaved stubble, where mineralisation was more intense, unlike with the undersown WM. This may have resulted in the higher nutrient availability for spring barley.

The literature presents different data on the effect of CC biomass on yield; increases of 14% [14], 7.9–22% [40], or 4.9% [34]. Ruark et al. [41] stated that Brassicaceae (radish) as a cover crop can result in neutral, negative, and positive effects on the subsequent crop yield. Most often, the increase in yield is associated with legume CCs [9,17]. Many studies have shown that cover crop nutrient release is influenced by the cover crop species and tillage treatments [21,30]. Due to more intense mineralisation, CC biomass ploughing-in results in a higher increase in plant yield compared to other incorporation methods [26,27,42]. Researchers have indicated that N taken up by brassica CCs is often not available when the subsequent crop needs it [41,43]. Yield is reduced due to the N deficit [9,17]. The reduced N available to plants can be attributed to N immobilisation by the soil microbial community. Sources in the literature indicate that there is a need to develop adaptive N fertilizer management that can overcome N immobilisation at key times of N demand [30,44]. Our studies show that N content accumulated in CC biomass and biomass C:N determined N uptake and other nutrients by the subsequent main crop (Figure 5).

Studies have shown that the transfer of nutrients from cover crops is not high (Table 4 and Table 5). Spring barley yields and N accumulated in them can be reduced by WM biomass with a C:N ratio > 20. This was also observed by other researchers [43]. In addition, the mineralisation of organic matter incorporated in heavy-textured soils is slower than in light-textured soils [6]. Overall, the positive effects of CCs on the physical, chemical, and biological soil properties are often associated with benefits (e.g., nutrient transfer) for the main crops [8].

## 3. Materials and Methods

### 3.1. Experimental Sites

The research was conducted in the northern part of Central Lithuania’s Lowland (56°12′ N, 24°20′ E) at the Joniškėlis Experimental Station of the Lithuanian Research Centre for Agriculture and Forestry. The soil of the experimental site is *Endocalcari-Endohypogleyic Cambisol* (siltic, drainic), whose texture is clay loam on silty clay with deeper lying sandy loam. The parent rock of this soil is limnoglacial clay, which at the depth of 70–80 cm transits into morenic loam. Clay particles < 0.002 mm in the Ap horizon (0–30 cm) account for 27.0%. The tests were conducted in the soil with the following agrochemical characteristics of the plough layer (0–25 cm): Experiment I: pH_KCl_ of 6.4, available P_2_O_5_ and K_2_O of 124–146 and 219–254 g kg^−1^, respectively, N_total_ of 0.15–0.17%, and SOC of 1.31–1.44% of soil; Experiment II: pH_KCl_ of 6.4, available P_2_O_5_ and K_2_O of 118–125 and 216–265 g kg^−1^, respectively, N_total_ of 0.14–0.17%, and SOC of 1.28–1.38% of soil; and Experiment III: pH_KCl_ of 6.4, available P_2_O_5_ and K_2_O of 183 and 268 g kg^−1^, respectively, N_total_ of 0.15%, and SOC of 1.62% of soil.

### 3.2. Meteorological Conditions

The meteorological conditions during both the main and post-harvest periods differed significantly among experiments (Table 6).

According to the data of Experiment I (2001–2004), the best conditions for CC growth were observed in 2001, characterised by a wet, warm, and long post-harvest period. The average daily temperature during this period was 0.6 °C higher compared to SCN. The worst conditions for CC growth were during the year 2002. The first half of the main growing season was dry, which resulted in poor emergence and development of RC. After the harvest, August was very dry and warm, which resulted in poor development of the undersown RC and slow germination of the post-sown WM. Only 74.6 mm of precipitation fell during the entire post-harvest period, 45% lower compared to the conditions in Experiment Ia. In addition, the average daily temperature dropped below +10 °C as early as 19 September, and in October, it was below +5 °C. The main growing period of the plants in 2003 was close to the standard climate normal (SCN). The month of July was slightly drier and hotter than SCN. More abundant rainfall only occurred at the end of August. CC germination was slow. The minimum daily temperature below +10 °C, recorded on 25 August, remained until the end of the CC growing season. We conclude that the conditions for CC growth were favourable in 2021 and 2003, and less favourable in 2002. The differences between the 2004 and 2006 growing seasons of Experiment II were even more pronounced. The main growing season in 2004 was cooler and drier compared to SCN. During this period, rainfall was 21.8% lower and the average daily temperature was 1.1 °C lower. August was characterised by an alternation of hot–dry and wet periods, with clayey soils becoming slushy and then drying out quickly. WM germinated slowly. In September, the minimum daily temperature rarely rose above +10 °C. In the second half of September, the average daily temperature was only +10 °C. The main growing season in 2006 was drier and warmer (temperature was 2.3 °C higher and rainfall was 74.3 mm lower) compared to 2004. There was a shortage of moisture in May–July. During this period, when cereals are growing intensively and drawing nutrients from the soil, only 38.9% of the rainfall occurred and temperatures were above SCN. The undersown red clover (RC) did not survive and was replaced by post-sown narrow-leaved lupins (NL). August was warm and rainy. Rainfall was twice as high compared to SCN, with most of it in the second ten-day period of the month. CC germinated and developed quickly and uniformly. Compared to SCN, the higher average daily temperatures persisted for the following two months, September and October. These meteorological data show that the main vegetation period and the yield of cultivated plants are also important for the growth of CCs, which can determine the amount of nutrients and moisture in the soil.

In 2013 (Experiment III), the main growing season was warmer (1.8 °C) with a similar, but unevenly distributed, rainfall compared to SCN. August was hot and dry. The main amount of precipitation for the post-harvest period occurred in September. Average daily temperatures below +10 °C only occurred during the ten-day period of September.

### 3.3. Experimental Designs and Details

Data from three experiments with CCs are presented and summarised in the article. Red clover and white mustard were grown as cover crops. Red clover (*Trifolium pratense* L.) is one of the many species belongings to the *Trifolium* L. genus, legume (Fabaceae) plant family, and white mustard (*Sinapis alba* L.) belongs to the genus of herbaceous plants of the Brassicaceae family. Cover crop species and their functions (Experiment I). The aim of the experiment was to determine the biomass and nutrient accumulation characteristics of CCs under different growing conditions. The research was conducted in the following plant sequence: winter wheat/spring barley and CC/spring barley. Three similar field experiments were set up in 2001–2002 (Ia), 2002–2003 (Ib), and 2003–2004 (Ic). The following experimental design was employed: Factor A: main crop: (1) winter wheat (*Triticum aestivum* L.) (WW); (2) spring barley (*Hordeum vulgare* L.) (SB). Factor B: cover crop: (1) without cover crop (WCC); (2) post-sown white mustard (*Sinapis alba* L.) (WM); (3) inter-sowing red clover (*Trifolium pratense* L.) (RC). Winter wheat (cultivar ‘Ada’, with a seed rate of 220 kg ha^−1^) and spring barley (cultivar ‘Ūla’, with a seed rate of 200 kg ha^−1^) were grown. Before sowing, the wheat and barley were fertilized with complex PK fertilizer (60 kg P ha^−1^ and 60 kg K ha^−1^). Ammonium nitrate (90 kg N ha^−1^) was applied to winter wheat at the beginning of the growing season and to barley (70 kg N ha^−1^) before sowing. The red clover (cultivar ‘Arimaičiai’, with a seed rate of 15 kg ha^−1^) was undersown in winter wheat in early spring as soon as the soil was workable, and in barley immediately after sowing. Cereal straw was collected and removed from the field. White mustard (cultivar ‘Braco’, with a seed rate of 15 kg ha^−1^) was sown on the same day after cereal harvesting. RC and WM were sown with special coulter attachments for the incorporation of fine seeds into the compacted soil.

Cover crop sowing methods and straw (Experiment II). The aim of this experiment was to clarify the efficiency of CCs with different sowing methods and their biomass incorporated together with straw. Two similar field experiments were conducted in 2004–2005 (IIa) and 2006–2007 (IIb) in a sequence of winter wheat + CC and spring barley. Experimental design: Factor A: winter wheat straw use patterns: (1) straw removed from the field (WS); (2) straw chopped and spread (+N) (S). Factor B: cover crops and their sowing methods: (1) without cover crops (WCCs); (2) post-sown white mustard (WMps); (3) undersown white mustard (WMus); (4) inter-sown red clover in 2005 (RC) or post-sown narrow-leaved lupine (*Lupinus angustifolius* L.) (NLps) in 2007. Red clover (cultivar ‘Vyliai’, with a seed rate of 15 kg ha^−1^) was undersown in winter wheat in early spring. After winter wheat harvesting, post-sown narrow-leaved lupin (2007; cultivar ‘Boruta’, with a seed rate of 180 kg ha^−1^) and WM (cultivar ‘Braco’, with a seed rate of 18 kg ha^−1^) were sown. The post-sown CCs were sown on the same day as the wheat was harvested and the straw and stubble were removed or chopped and spread (on 3 August). The inter-sown WM was undersown before winter wheat harvesting (at the early waxy maturity stage, on 24 July) with a disc fertilizer spreader, after increasing the seed rate to 25 kg ha^−1^. Ammonium nitrate (+45 kg N ha^−1^) was applied for straw mineralisation, except in the plots with legumes (RC or LN).

Fertilization systems (Experiment III). The aim of the experiment was to evaluate the effectiveness of CCs in crop rotations with different fertilization intensities (mineral fertilizers and straw). The experiment was established in 2013. Spring barley cultivar ‘Noja DS’ was seeded at a rate of 220 kg ha^−1^. The following experimental design was employed: Factor A: cover crop: (1) without a cover crop (WCC); (2) post-sown white mustard (WM); (3) inter-sown red clover (RC). Factor B: fertilization of spring barley: (1) unfertilized (UF); (2) sustainable fertilizing (SF); (3) intensive fertilizing (IF). In the CC treatment with RC, the variety ‘Sadūnai’ was undersown in spring barley at a seed rate of 10 kg ha^−1^. In the treatment with WM, the variety ‘Braco’ was sown shortly after spring barley harvesting and stubble tillage at a seed rate of 15 kg ha^−1^. Mineral fertilizer rates were calculated considering the agrochemical indicators of the soil and using recommendations for the fertilizer rate calculation to achieve a target spring barley yield. A moderate spring barley yield was targeted (4.0 t ha^−1^) with a moderate fertilization (72 kg N ha^−1^) rate (SF) and a high yield (5.5 t ha^−1^) with a high fertilization (108 kg N ha^−1^) rate (IF). When spring barley was grown with undersown RC, the nitrogen fertilizer rate was reduced by 20%. In the field experiment, the crops were grown according to conventional farming standards.

In mid-October, all CCs were rolled, incorporated with disc cultivators and then ploughed (at a 25 cm depth). After the incorporation of the CC biomass, spring barley was grown the following year according to conventional farming standards, while applying N_60_, P_60_, and K_60_ fertilizers (Experiments I and II). The experimental plots were laid out in a complete two-factor randomised block design with four replicates. The individual plot size was 5 × 20 m.

### 3.4. Plant and Soil Analyses

The shoot biomass of CCs before their ploughing-in (in the first part of October) was measured. The sampling of the shoot biomass of four randomly chosen squares of 0.25 m^2^ in each plot was cut to ground level and weighed. After plant mass weighing, the dry matter was determined (dried to constant mass at 105 °C) and the shoot biomass of RC and WM was calculated. Winter wheat and spring barley were harvested when the majority of plants reached the hard dough stage (BBCH 87) with a small-size combination harvester. Straw and grain yield was measured by weight. Spring barley grain and straw samples were taken from each plot for the determination of the DM content and chemical composition. All samples (CCs, cereals grain, and straw) were dried and ground using a ZM200 ultra-centrifugal mill (Retch, Haan, Germany) with 1 mm mesh sieves and the N, P, K, and C (for CC) contents were analysed. The concentrations of N, P, and K were evaluated in sulphuric acid digestates. The samples for N_tot_ determination were analysed using the Kjeldahl method with a Kjeltec system 1002 (Foss Tecator, Höganäs, Sweden) [45] The concentration of P was quantified spectrophotometrically by a coloured reaction with ammonium *molybdate-vanadate* at a wavelength of 430 nm on a spectrophotometer Cary 50 UV-Vis (Varian Inc., Palo Alto, CA, USA). The respective K concentration was evaluated by atomic absorption spectrometry with an AAnalyst 200 (Perkin Elmer, Waltham, MA, USA) in accordance with the manufacturer’s instructions. The concentrations of C in the CC samples were determined by a spectrophotometric procedure at a 590 nm wavelength after wet oxidation with potassium dichromate and sulphuric acid solution. The soil mineral nitrogen (SMN = N-NO_3_ + N-NH_4_) was measured 2 times during the experimental period: in autumn before the ploughing-in of CCs and in spring before the barley was sown. Nitrate nitrogen (N-NO_3_) was determined by the ionometric method and ammonium nitrogen (N-NH_4_) by the spectrophotometric method.

### 3.5. Statistical Analysis

Statistical analysis was performed using the software package SELEKCIJA [46]. Collected data were subjected to a three-way analysis of variance (ANOVA). The procedure was performed considering the factors: experiments/years, cover crop species and their sowing methods, and the fertilization of the main crops and straw use. Significant differences between factors were determined by an F-test at the *p* < 0.05 and *p* < 0.01 probability levels. The factors marked as significant (by the *p*-value) and interactions were analysed by the difference from the check using the least significant difference (LSD_05_). Significant differences in data for the NPK of spring barley were calculated by a Duncan’s test (a one-way ANOVA) at *p* < 0.05, where the means with the same letter are not significantly different (Experiment III). The standard error of the mean *(SE*) was used to represent the error values and bars. The relationships among the experimental data were investigated using a linear regression analysis with the software STAT ENG, version 1.5.

## 4. Conclusions

We conclude that the ground mass of intermediate crops was positively influenced by agronomic practices, such as the cultivation of WM and RC in the field of short-vegetation spring barley, less intensive (SF) fertilization of the main crop, the use of straw as fertilizer (+N), and the inland CC sowing method. However, in most cases, the studied measures were in interaction with the meteorological conditions. The plants (RC and WMus) were negatively affected by the prolonged dry period after sowing.

For post-sown plants (WMps), not only the amount of moisture during germination is important, but also the average daily temperature of the entire growth period. Nutrient (NPK) accumulation depended on TP biomass. RC yields were negatively affected by intensive fertilization of main crops and straw fertilization. On the contrary, this agronomic practice and growing WM in the summer field increased the productivity of WM. As the amount of N accumulated in CC biomass increased, SMN decreased. The research data showed that RC biomass has a higher N concentration, while WP has a higher P concentration. The biomass mineralization ratio (C:N) has a more considerable variation in WM biomass (10–27) than in RC (9–16). CC biomass, N, and C:N had a significant influence on the SMN content in spring and NPK accumulation in the yield of subsequent crops.

## Figures and Tables

**Figure 1 plants-12-02966-f001:**
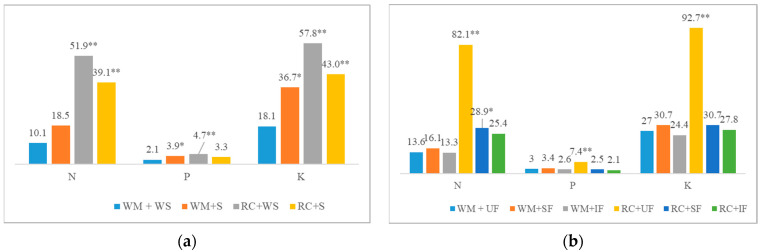
Effect of the interaction of cover crop species with straw fertilization (**a**) and the fertilization of main crops (**b**) on nutrient uptake. Values with asterisks indicate significant differences from the control (* *p* ≤ 0.05; ** *p* ≤ 0.01), based on Fisher’s LSD test.

**Figure 2 plants-12-02966-f002:**
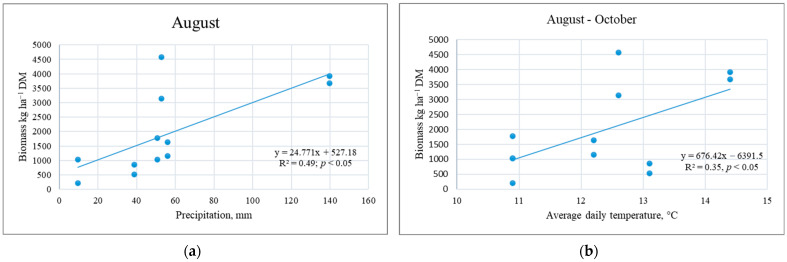
Dependence of the shoot biomass of white mustard cover crops on rainfall (**a**) and the daily mean temperature (**b**).

**Figure 3 plants-12-02966-f003:**
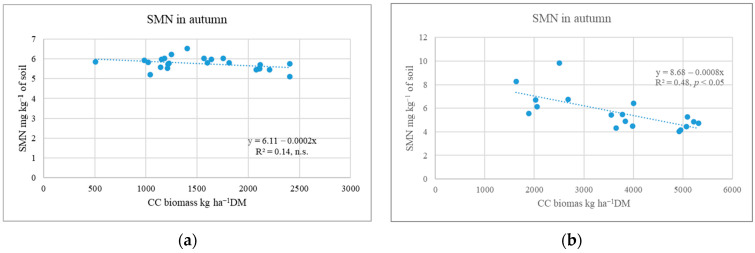
Dependence of soil mineral N on CC shoot biomass during different post-harvest periods (Experiment II). (**a**) Post-harvest period not very favourable for CC growth; (**b**) post-harvest period favourable for CC growth.

**Figure 4 plants-12-02966-f004:**
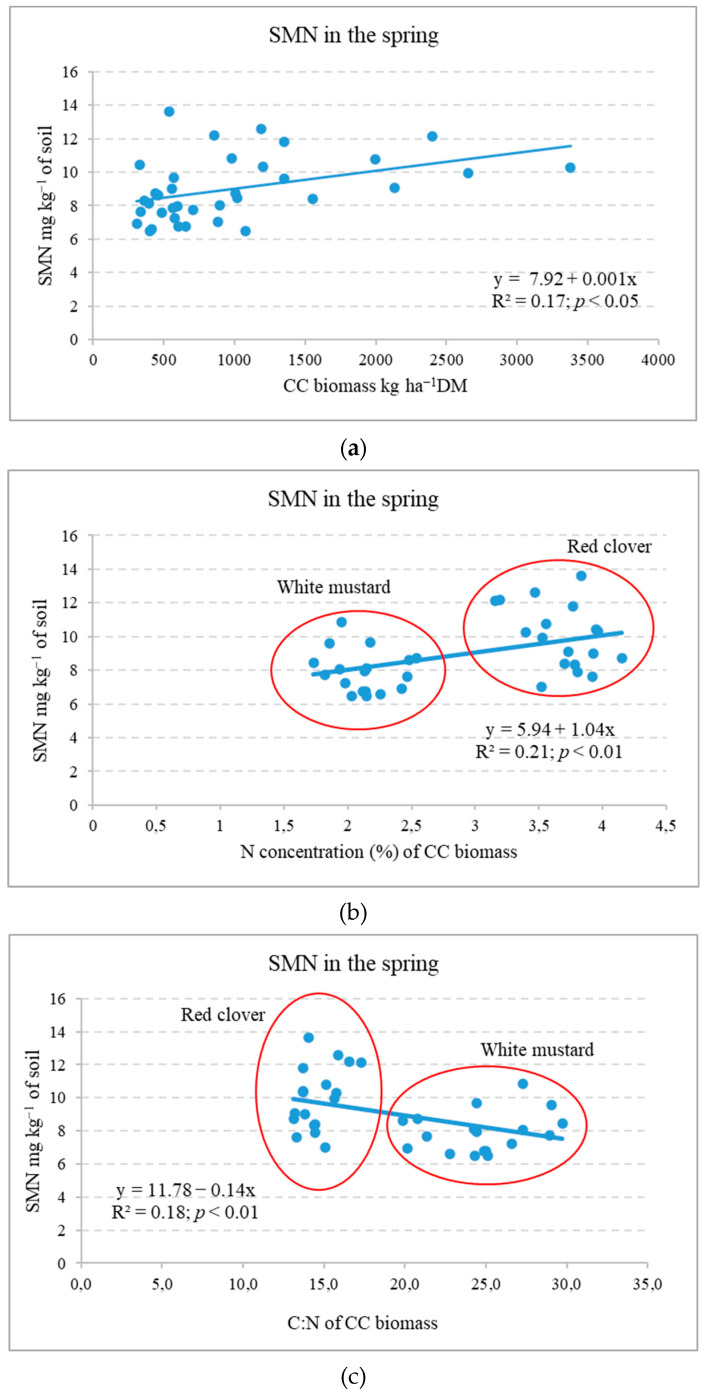
Dependence of the soil mineral N on the incorporated cover crop shoot biomass (**a**) and its N concentration (**b**) and C:N (**c**) in spring (Experiment III).

**Figure 5 plants-12-02966-f005:**
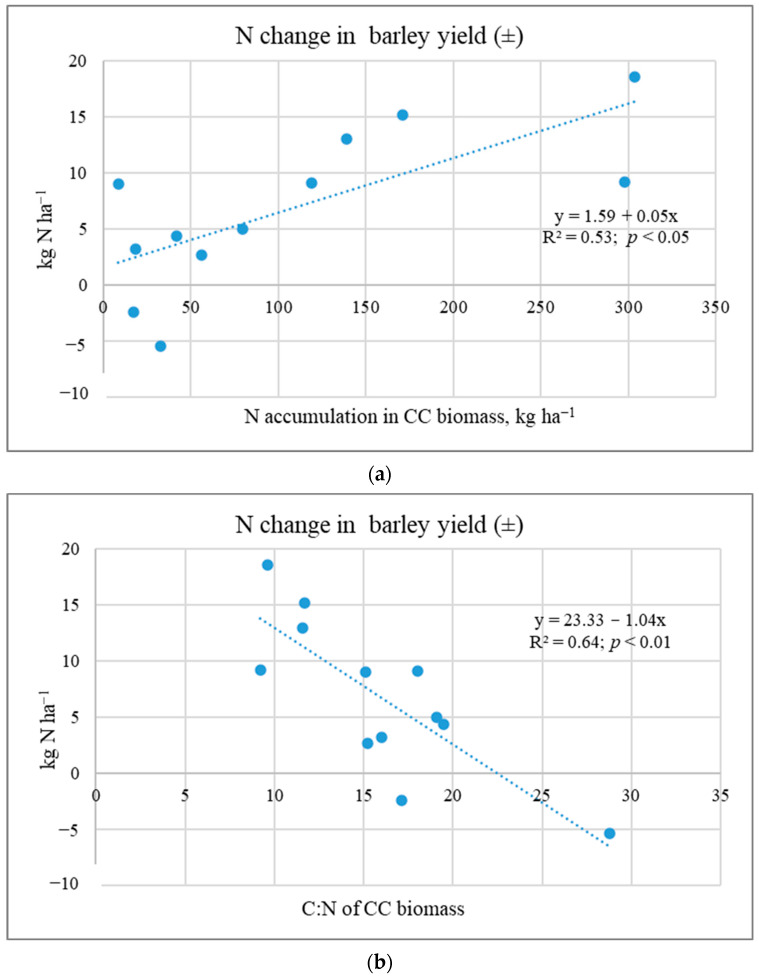
Dependence of N accumulation (**a**) in spring barley yields on CC biomass qualitative parameters C:N (**b**) (Experiment I).

**Table 1 plants-12-02966-t001:** Shoot biomass and nutrient accumulation of different types of cover crops (mean ± SE).

Experiment (E)	Main Crop (MC)	Cover Crop (CC)	Biomass	N	P	K
			kg ha^−1^ DM			
Ia	WW	WM	3141 ± 183	72.8 ± 6.4	3.7 ± 0.3	65.3 ± 3.9
		RC	6050 ± 185 **	184.9 ± 12.3 **	31.4 ± 0.7 **	199.0 ± 5.8 **
	SB	WM	4580 ± 131 **	111.7 ± 2.6 **	6.4 ± 0.7	101.0 ± 7.6 **
		RC	6590 ± 151 **	250.8 ± 13.7 **	30.3 ± 0.6 **	276.1 ± 5.8 **
Ib	WW	WM	1032 ± 190 **	16.4 ± 4.4 **	2.8 ± 0.5	21.2 ± 3.7 **
		RC	348 ± 66 **	9.3 ± 1.9 **	0.8 ± 0.1	7.6 ± 1.5 **
	SB	WM	210 ± 21 **	5.8 ± 0.2 **	0.5 ± 0.1 *	3.8 ± 0.5 **
		RC	220 ± 18 **	6.3 ± 0.6 **	0.4 * ± 0.0	7.2 ± 0.7 **
Ic	WW	WM	1030 ± 188 **	20.2 ± 4.1 **	2.0 ± 0.3	21.3 ± 3.8 **
		RC	3020 ± 740	86.2 ± 22.2	11.4 ± 3.3 **	82.5 ± 19.5
	SB	WM	1782 ± 70 **	46.9 ± 2.7	3.6 ± 0.1	48.7 ± 2.7
		RC	3275 ± 57	117.2 ± 2.3 **	10.5 ± 0.2 **	98.6 ± 1.6 **
Probability (*p*) level of the factors and their interaction						
E (2)			<0.001	<0.001	<0.001	<0.001
MC (1)			<0.05	<0.001	ns	<0.001
CC (1)			<0.001	<0.001	<0.001	<0.001
ExMC (2)			<0.01	0.001	ns	<0.001
ExCC (2)			<0.001	<0.001	<0.001	<0.001
MCxCC (1)			ns	ns	ns	ns
ExMCxCC (2)			ns	ns	ns	<0.05

Note. WM, white mustard; RC, red clover; WW, winter wheat; SB, spring barley. Values with asterisks indicate significant differences from the control (* *p* ≤ 0.05; ** *p* ≤ 0.01), based on Fisher’s LSD test. SE, standard error; (1), (2), degrees of freedom, ns, not significant.

**Table 2 plants-12-02966-t002:** Influence of the sowing methods and cereal straw use on the CC shoot biomass and nutrient accumulation (mean ± SE).

Experiment (E)	Straw Use (SU)	Cover Crops and Sowing Methods (CC)	Biomass	N	P	K
			kg ha^−1^ DM			
IIa	WS	WMps	1150 ± 68	38.9 ± 2.4	6.7 ± 0.5	34.8 ± 2.1
		WMus	1040 ± 272	38.7 ± 10.1	6.5 ± 1.6	28.8 ± 7.7
		RCus	2100 ± 171 **	60.4 ± 3.5 **	6.9 ± 0.5	50.0 ± 4.2 *
	S	WMps	1640 ± 58 *	53.1 ± 1.7 *	8.4 ± 0.3	51.3 ± 1.9 *
		WMus	1140 ± 79	34.0 ± 2.3	4.7 ± 0.3 *	26.0 ± 1.7
		RCus	2210 ± 98 **	62.7 ± 2.3 **	7.3 ± 0.6	44.8 ± 1.8
IIb	WS	WMps	3680 ± 83 **	103.7 ± 0.3 **	15.4 ± 0.5 **	149.8 ± 4.7 **
		WMus	5070 ± 87 **	152.6 ± 3.5 **	19.2 ± 0.3 **	168.8 ± 1.2 **
		NLps	1990 ± 50 **	58.3 ± 1.7 **	6.6 ± 0.4	49.8 ± 1.3 *
	S	WMps	3920 ± 72 **	152.5 ± 3.4 **	16.2 ± 0.4 **	142.7 ± 3.3 **
		WMus	5119 ± 105 **	201.8 ± 5.8 **	20.2 ± 0.4 **	189.0 ± 6.7 **
		NLps	2270 ± 324 **	66.6 ± 9.2 **	7.2 ± 1.1	50.4 ± 8.4 *
Probability (*p*) level of the factors and their interaction						
E (1)			<0.001	<0.001	<0.001	<0.001
SU (1)			<0.05	<0.001	ns	ns
CC (2)			<0.001	<0.001	<0.001	<0.001
ExSU (1)			ns	<0.001	ns	ns
ExCC (2)			<0.001	<0.001	<0.001	<0.001
SUxCC (2)			ns	<0.01	ns	ns
ExSUxCC (2)			ns	<0.01	ns	<0.01

Note. WS, straw removed from the field; S, straw chopped and spread; WMps, post-sown white mustard; WMus, undersown white mustard; RCus, undersown red clover; NLps, post-sown narrow-leaved lupine. Values with asterisks indicate significant differences from the control (* *p* ≤ 0.05; ** *p* ≤ 0.01), based on Fisher’s LSD test. SE, standard error; (1),(2) degrees of freedom; ns, not significant.

**Table 3 plants-12-02966-t003:** Effect of fertilizer intensity on shoot biomass and nutrient concentration in cover crop biomass (mean ± SE).

Cover Crops (CCs)	Straw Use (SU)	Fertilization (F)	Biomass kg ha^−1^ DM	N	P	K
				% DM		
WM	WS	UF	524 ± 63	2.04 ± 0.04	0.45 ± 0.01	3.91 ± 0.06
		SF	460 ± 103	2.27 ± 0.09	0.48 ± 0.03	3.81 ± 0.17
		IF	397 ± 36	2.37 ± 0.11	0.48 ± 0.00	4.12 ± 0.03
	S	UF	858 ± 144	1.95 ± 0.13	0.44 ± 0.02	3.96 ± 016
		SF	1107 ± 133 *	1.98 ± 0.09	0.42 ± 0.01	3.95 ± 0.10
		IF	773 ± 122	2.17 ± 0.21	0.43 ± 0.02	4.18 ± 0.12
RC	WS	UF	2445 ± 527 **	3.42 ± 0.16 **	0.32 ± 0.04 **	3.90 ± 0.26
		SF	1131 ± 146 *	3.48 ± 0.17 **	0.31 ± 0.01 **	3.75 ± 0.07
		IF	873 ± 193	3.77 ± 0.13 **	0.31 ± 0.00 **	4.14 ± 0.11
	S	UF	2263 ± 200 **	3.61 ± 0.06 **	0.30 ± 0.00 **	3.99 ± 0.17
		SF	458 ± 67	3.93 ± 0.01 **	0.34 ± 0.00 **	4.06 ± 0.13
		IF	455 ± 60	3.91 ± 0.12 **	0.32 ± 0.01 **	4.21 ± 0.08
Probability (*p*) level of the factors and their interaction						
CC (1)			<0.001	<0.001	<0.001	ns
SU (1)			ns	ns	ns	ns
F (2)			<0.001	<0.01	ns	<0.05
CCxSU (1)			0.001	<0.01	<0.05	ns
CCxF (2)			<0.001	ns	ns	ns
SUxF (2)			ns	ns	ns	ns
CCxSUxF (2)			ns	ns	ns	ns

Note. WM, white mustard; RC, red clover; UF, not fertilized; SF, sustainable fertilizer; IF, intensive fertilizer; WS, straw removed from the field; S, straw chopped and spread. Values with asterisks indicate significant differences from the control (* *p* ≤ 0.05; ** *p* ≤ 0.01), based on Fisher’s LSD test. SE, standard error; (1), (2) degrees of freedom; ns, not significant.

**Table 4 plants-12-02966-t004:** Influence of cover crops on nutrient accumulation in spring barley yields (mean ± SE).

Experiment (E)	Main Crop (MC)	Cover Crop (CC)	N	P	K
			kg ha^−1^		
Ia	WW	WCC	74.4 ± 0.9 a	17.2 ± 0.3 a	81.5 ± 1.0 a
		WM	79.4 ± 1.4 b	19.2 ± 0.3 b	80.8 ± 1.0 a
		RC	83.6 ± 1.6 c	19.0 ± 0.5 b	89.3 ± 1.0 b
	SB	WCC	96.4 ± 1.1 d	20.9 ± 0.4 c	86.0 ± 0.3 ab
		WM	105.5 ± 0.8 e	22.0 ± 0.1 d	90.3 ± 0.2 b
		RC	115.0 ± 0.5 f	22.8 ± 0.1 d	94.4 ± 0.3 c
Ib	WW	WCC	102.5 ± 0.9 b	21.1 ± 1.3 b	97.5 ± 0.4 ab
		WM	97.1 ± 1.5 ab	19.4 ± 0.4 a	94.7 ± 0.6 a
		RC	100.1 ± 1.2 b	20.4 ± 0.1 a	99.0 ± 0.5 b
	SB	WCC	95.4 ± 0.8 a	19.3 ± 0.1 a	94.8 ± 1.0 a
		WM	104.4 ± 0.1 b	21.4 ± 0.1 b	97.3 ± 0.2 ab
		RC	98.6 ± 1.9 ab	19.9 ± 0.9 a	97.3 ± 0.2 ab
Ic	WW	WCC	108.0 ± 1.0 ab	25.5 ± 0.3 b	98.0 ± 0.5 a
		WM	112.4 ± 1.3 b	26.5 ± 0.4 b	99.5 ± 0.4 b
		RC	121.0 ± 0.9 c	27.7 ± 0.4 c	102.6 ± 1.0 c
	SB	WCC	101.5 ± 0.7 a	23.5 ± 0.7 a	98.3 ± 0.5 ab
		WM	103.7 ± 3.6 a	24.1 ± 0.9 a	98.3 ± 1.0 ab
		RC	116.2 ± 0.2 b	25.7 ± 0.3 b	104.3 ± 0.5 c
Probability (*p*) level of the factors and their interaction					
E (2)			<0.001	<0.001	<0.001
MC (1)			<0.001	ns	<0.001
CC (2)			<0.001	<0.001	<0.001
ExMC (2)			<0.001	<0.001	<0.001
ExCC (4)			<0.001	<0.05	<0.001
MCxCC (2)			<0.001	ns	<0.001
ExMCxCC (4)			<0.001	<0.05	<0.001

Note. WM, white mustard; RC, red clover; WW, winter wheat; SB, spring barley. Values followed by different letters are significant differences (*p* ≤ 0.05), based on Fisher’s LSD test. SE, standard error; (1), (2), (4) degrees of freedom; ns, not significant.

**Table 5 plants-12-02966-t005:** Influence of cover crops, sowing methods, and straw use on the nutrient content of spring barley yields (mean ± SE).

Experiment (E)	Straw Use (SU)	Cover Crop (CC) and Sowing Methods	N	P	K
			kg ha^−1^		
IIa	WS	WCC	74.2 ± 0.9 a	11.7 ± 0.4 b	54.5 ± 0.9 d
		WMps	86.1 ± 1.1 c	12.9 ± 0.2 d	59.7 ± 0.9 e
		WMus	84.5 ± 4.2 c	12.7 ± 0.7 cd	54.2 ± 0.3 cd
		RCus	77.6 ± 3.4 a	12.0 ± 0.6 bcd	65.2 ± 1.1 f
	S	WCC	76.0 ± 5.2 a	11.4 ± 0.3 b	43.5 ± 0.4 a
		WMps	78.2 ± 1.4 a	11.5 ± 0.5 b	52.2 ± 0.9 c
		WMus	73.9 ± 1.0 a	10.2 ± 0.5 a	47.6 ± 1.3 b
		RCus	75.9 ± 1.8 a	11.6 ± 0.4 b	45.7 ± 0.5 b
IIb	WS	WCC	92.7 ± 1.7 ab	15.8 ± 0.2 a	76.9 ± 1.2 a
		WMps	100.4 ± 2.9 cde	16.9 ± 0.5 bc	93.8 ± 1.1 cd
		WMus	103.0 ± 2.6 e	17.0 ± 0.3 c	88.7 ± 0.4 bcd
		RCus	98.7 ± 1.8 cde	16.5 ± 0.4 abc	93.9 ± 3.5 cd
	S	WCC	91.0 ± 0.9 a	16.2 ± 0.3 abc	83.0 ± 2.6 ab
		WMps	98.2 ± 0.9 cde	16.4 ± 0.3 abc	93.0 ± 3.7 cd
		WMus	98.9 ± 5.3 cde	16.9 ± 0.6 bc	84.7 ± 0.9 b
		RCus	96.4 ± 1.1 bc	16.3 ± 0.2 abc	93.9 ± 2.2 d
Probability (*p*) level of the factors and their interaction					
E (1)			<0.001	<0.001	<0.001
SU (1)			<0.05	<0.01	<0.001
CC (3)			<0.01	ns	<0.001
ExSU (1)			ns	<0.05	<0.001
ExCC (3)			ns	ns	<0.05
SUxCC (3)			ns	ns	<0.05
ExSUxCC (3)			ns	ns	<0.01

Note. WS, straw removed from the field; S, straw chopped and spread; WMps, post-sown white mustard; WMus, undersown white mustard; RCus, undersown red clover; NLps, post-sown narrow-leaved lupine. Values followed by different letters are significant differences (*p* ≤ 0.05), based on Fisher’s LSD test. SE, standard error; (1), (3) degrees of freedom; ns, not significant.

**Table 6 plants-12-02966-t006:** Meteorological conditions during the growing seasons of plants at the Joniškėlis Experimental Station of the Lithuanian Research Centre for Agriculture and Forestry.

Month		2001	2002	2003	2004	2006	2013	SCN
April	T, °C	7.6	6.9	4.2	6.4	6.7	4.6	6.2
	P, mm	36.4	28.4	37.1	8.3	23.3	36.9	37.4
May	T, °C	10.5	14.6	12.3	10.3	12.2	16.2	12.3
	P, mm	35.4	10.9	56.1	25.2	32.5	73.4	45.6
Junie	T, °C	13.1	16.4	14.1	13.8	16.3	18.6	15.6
	P, mm	147.9	81.1	57.2	64.0	6.8	44.4	59.4
July	T, °C	19.6	19.8	19.7	16.1	20.9	18.9	17.2
	P, mm	172.5	41.5	45.1	68.0	28.6	53.0	69.2
Main period	T, °C	12.7	14.4	12.6	11.7	14.0	14.6	12.8
	P, mm	392.2	161.9	195.5	165.5	91.2	207.7	211.6
August	T, °C	16.2	18.1	16.5	17.4	18.7	18.6	17.1
	P, mm	52.8	9.6	50.6	56.0	139.8	38.8	67.9
September	T, °C	11.1	12,4	12.2	12.3	15.0	12.4	12.0
	P, mm	59.8	27.5	29.5	69.8	48.2	81.1	57.9
October	T, °C	9.9	2.4	6.1	4.4	9.4	6.3	6.3
Until 15	P, mm	22.7	37.5	37.2	9.0	48.5	0.2	45.5
Post-harvest period	T, °C	12.4	11.0	11.6	11.4	14.4	12.4	11.8
	P, mm	135.3	74.6	117.3	134.8	236.5	120.1	161.3

Note. T, mean daily air temperature; P, sum of the precipitation; SCN, standard climate normal; post-harvest period, August–October; main period, April–July.

## Data Availability

The datasets used and/or analysed during the present study are available from the corresponding author upon reasonable request.

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
