# Peer review of "Cover Crop Yield, Nutrient Storage and Release under Different Cropping Technologies in the Sustainable Agrosystems"

_plants, 2023, doi:10.3390/plants12162966_

Round 1
Reviewer 1 Report
In this study, the authors describe different cover crop yield, nutrient storage and release under different cropping technologies. The manuscript is interesting, well-written and the study is well-performed, but there are some corrections necessary:
#In the abstract, please make it clear that RC is red clover and WM is white mustard. Why WW (winter wheat) and SB (spring barley) are not commented in the abstract?
#Tables. Mean results should accompany the confidence interval (CI95%).
#Also in Tables. Please change the p-values below 0.001 from 0.000 to the lowest decimal place, for example, if the p = 0.0003, p = 10^-4.
#What is "ns" in the Tables? "non-significant".
#Why have the authors adopted the Fisher's LSD test to calculate the p-values?
#Please add p-values in Fig. 2, as you did in Fig.3.
#Section 3.3. Something the authors could use in future studies is Design of Experiments (DoE), a mathematical way to vary all the parameters (including fertilization and weather) at the same time to extract the global optimum (SU and CC that will provide [considering changes of fertilizers and weather conditions] the best nutrient content, etc, depending on the measured response).
#In section 3.4, please add references for the procedures undertaken to measure CC, and nutrients (N, P and K).
Author Response
Dear Collegue,
Thank you for your comments, recommendations and time for review.
Kind regards,
Authors

Reviewer 2 Report
The manuscript tilted" Cover crop yield, nutrient storage and release under different cropping technologies in the sustainable agrosystems". Actually, the idea of the article is not new but it has very good and planed work. I have some comments on this manuscript; these comments could help the author to represent their work more good.
Abstract:
Authors mentioned only two crops but they did not mentioned anything about the wheat or barely. Please add one sentence on the two crops, which will make the reader impressed from the work.
In the introduction;
Authors talked about biodiversity, sustainability and cover crops; please add definition for the cover crop in in the begging of line 51, page 2.
Soil biodiversity, it plays an important role in cover crop, I hope if the authors listed a small paragraph on the important of soil microorganisms in cover crop and crop yield.
Authors in line 110 page 3, said fertilization and in the abstract write it as fertilization (line 20), please used unique word.
References and citation is low, please add more recent references in the introduction.
Results and discussion
Number of table is too much, could the authors reduce to four because we have 4 figures.
Interpretation of the results when combined with discussion should be better than what I read. Please used more citation to represent your results and show the how much is valuable in compared with the others.
Figure 5 should be more conspicuous because its color is pale.
Materials and methods
Are written very well and I have no comment on it.
Conclusion should be separated and contains a convey, because this article is so important for the farmer, authors should summarized a good conclusion in very simple manner.
Author Response

(The authors gave the same response as above.)
